# Efficient simulation of Time-Fractional Korteweg-de Vries equation via conformable-Caputo non-Polynomial spline method

**Majeed A. Yousif** [1]\*, **Faraidun K. Hamasalh** [2], **Ahmad Zeeshan** [3], **Mohamed Abdelwahed** [4]

**1** Department of Mathematics, College of Education, University of Zakho, Duhok, Iraq, **2** Department of Mathematics, College of Education, University of Sulaimani, Sulaimani, Iraq, **3** Department of Mathematics and Statistics, FBAS, International Islamic University Islamabad, Islamabad, Pakistan, **4** Department of Mathematics, College of Sciences, King Saud University, Riyadh, Saudi Arabia

\* majeed.yousif@uoz.edu.krd

## Abstract

This research presents a novel conformable-Caputo fractional non-polynomial spline method for solving the time-fractional Korteweg-de Vries (KdV) equation. Emphasizing numerical analysis and algorithm development, the method offers enhanced precision and modeling capabilities. Evaluation via the Von Neumann method demonstrates unconditional stability within defined parameters. Comparative analysis, supported by contour and 2D/3D graphs, validates the method's accuracy and efficiency against existing approaches. Quantitative assessment using $L_2$ and $L_\infty$ error norms confirms its superiority. In conclusion, the study proposes a robust solution for the time-fractional KdV equation.

**Data Availability Statement:** All relevant data are within the paper.

## 1 Introduction

Nonlinear fractional differential equations (FDEs) occupy a central position in the landscape of mathematical challenges, permeating a multitude of disciplines such as physics, chemistry, engineering, and finance [1–10]. Their applications transcend theoretical realms, finding practical utility in modeling physical systems and predicting trends in financial markets, showcasing the profound impact these equations have on our understanding of complex phenomena. Researchers, recognizing the intricate behavior that often eludes traditional analytical techniques, have immersed themselves in the exploration and resolution of nonlinear time-fractional differential equations. This pursuit has given rise to a rich tapestry of numerical methods, each offering unique approaches to solving these intricate problems. Techniques such as B-spline collocation [11], polynomial and non-polynomial splines [12], Galerkin–Hermite methods [13], and integro-quadratic spline-based schemes [14] serve as practical avenues, providing researchers with tools to approximate solutions tailored to the specific equation at hand and the desired level of accuracy. Among the prominent nonlinear fractional differential equations, the time-fractional Korteweg-de Vries equation, formulated by Dutch mathematicians in 1895, stands out as a captivating focal point [15]. Its applications extend beyond its

**Funding:** Researchers Supporting Project number (RSP2024R136), King Saud University, Riyadh, Saudi Arabia.

**Competing interests:** The authors have declared that no competing interests exist.

historical roots, encompassing the study of wave phenomena in diverse physical systems, including shallow water wave propagation and rogue wave dynamics. The fractional generalization of the KdV equation amplifies its capacity to represent anomalous diffusion and dispersion, prompting the utilization of various numerical methods to unravel its complexities and facilitate the study of nonlinear waves in various contexts. These methods include the homotopy analysis transform method [16], Green's Function with the Mittag-Leffler function [17], and the Radial Basis Functions Collocation Method [18]. Additionally, recent works have explored alternative numerical approaches such as the Laplace transform decomposition method [19], finite element method with B-spline [20], spectral collocation method with Chebyshev polynomials [21], finite difference/collocation scheme based on Jacobi–Gauss–Lobatto nodes [22], and q-homotopy analysis transform method [23]. These methods collectively contribute to the comprehensive understanding and analysis of the time-fractional KdV equation, offering valuable insights into its behavior and applications across diverse disciplines. Amid the vast array of numerical methods, the non-polynomial spline method emerges as a versatile and indispensable tool, showcasing its efficacy in navigating a diverse spectrum of mathematical challenges [24–26]. Its adaptability shines through in applications ranging from standard differential equations to fractional differential equations and nonlinear Volterra integral equations. The method's versatility is exemplified in its successful resolution of challenges such as the fractional Bagely-Torvik equation and the Burgers-Fisher fractional differential equation [27–34]. The burgeoning interest evident in the surge of studies dedicated to non-polynomial spline methods underscores their importance and solidifies their role in propelling the field of mathematical research forward. Contemplate the time-fractional Korteweg-de Vries equation with the nonlinear term [35]:

$$\frac{\partial^\alpha u(x,t)}{\partial t^\alpha} + \frac{\partial^3 u(x,t)}{\partial x^3} + \sigma_1 u(x,t)\frac{\partial u(x,t)}{\partial x} = F(x,t), \quad 0 < \alpha \le 1, \tag{1}$$

with boundary conditions at $t \ge 0$, as follows:

$$\begin{aligned} u(a,t) &= \phi_1(t), \\ u(b,t) &= \phi_2(t), \\ u_x(b,t) &= \phi_3(t), \end{aligned} \tag{2}$$

and the initial condition:

$$u(x,0) = \Psi(x) \quad x \in [a,b]. \tag{3}$$

where $\sigma_1$ is a specified constant and Caputo fractional derivative of the function $u(x,t)$ is defined as [36]:

$$\frac{\partial^\alpha u(x,t)}{\partial t^\alpha} = D_t^\alpha(u(x,t)) = \left\{ \begin{array}{ll} \frac{1}{\Gamma(n-\alpha)} \int_0^t \frac{u^{(n)}(x,s)}{(t-s)^{\alpha+1-n}}\ ds, & n-1 < \alpha \le n, \\ \frac{\partial u^n(x,t)}{\partial t^n}, & \alpha = n. \end{array} \right\}. \tag{4}$$

The time-fractional Korteweg-de Vries equation, given by Eq (1), arises from the combination of physical principles and mathematical considerations relevant to the behavior of nonlinear waves in one-dimensional media, such as shallow water waves and plasma waves.

This research responds to the critical need for more effective numerical methodologies to tackle the time-fractional Korteweg-de Vries Eqs (1)–(3) and its counterparts in fluid mechanics, plasma physics, and mathematical physics. Existing methods often struggle to capture the intricate dynamics and complex behavior inherent in these equations. To address this challenge, we propose the conformable-Caputo fractional non-polynomial spline method, a novel

approach that embeds fractional derivatives within a non-polynomial spline framework. This integration significantly enhances accuracy and modeling capabilities, offering a more precise approximation of the solution compared to traditional polynomial-based methods. Our study not only introduces this innovative method but also demonstrates its stability and efficiency through rigorous analysis and comparisons with existing approaches. We employ stability analysis techniques, such as the von Neumann method, to provide insights into the method's robustness within specific parameter ranges, ensuring reliable and consistent results. While our focus is on numerical aspects, future research avenues include exploring the method's broader applicability and theoretical underpinnings. The key advantages of our approach over traditional methods lie in its ability to handle memory effects, irregular data, and systems with fractional dynamics more effectively, making it a versatile and accurate tool for modeling across various scientific domains. Our ultimate goal is to showcase the potential of this innovative method not only for the KdV equation but also in applied fields such as fluid mechanics, plasma physics, oceanography, geophysics, and biology.

## 2 Caputo fractional non-Polynomial spline construction

Within this section, our focus has been on the construction of a fractional non-polynomial spline, a novel mathematical entity that incorporates not only polynomial elements but also features trigonometric and exponential functions. This departure from the traditional spline framework introduces fractional calculus into the mix, allowing for a more refined and nuanced representation of mathematical structures. By integrating trigonometric and exponential functions within the context of fractional non-polynomial splines, we aim to harness the unique characteristics of these functions to provide a more flexible and expressive tool for addressing fractional differential equations. This innovative construction offers a promising avenue for exploring the complex dynamics associated with fractional calculus, paving the way for enhanced precision and adaptability in solving mathematical problems that exhibit fractional behavior.

Consider $x_j$ defined as $j(h)$ for $j = 0, 1, \cdots, M$, and $t_n$ defined as $n(\tau)$ for $n = 0, 1, \cdots, N$, where $h = \frac{b-a}{M}$ represents the uniform spatial interval and $\tau = \frac{T}{N}$ denotes the uniform temporal interval. To formulate a numerical strategy for simulating the solution to Eq (1), we will employ a fractional non-polynomial spline.

$$
\begin{aligned}
S_{j,n}(x_j, t_n) &= a_j^n \cos k(x - x_j) + b_j^n \sin k(x - x_j) \\
&\quad + c_j^n e^{k(x-x_j)} + d_j^n (x - x_j) + e_j^n.
\end{aligned}
\tag{5}
$$

Let $S_{j,n}(x_j, t_n)$ denote the non-polynomial spline function, approximating the solution $u_j^n = u(x_j, t_n)$. In this context, $k$ signifies the frequency for trigonometric functions, while the coefficients $a_j^n, b_j^n, c_j^n, d_j^n$, and $e_j^n$ remain undetermined. The determination of these coefficients relies on the fulfillment of the ensuing conditions:

$$
\begin{aligned}
S_{j,n}(x_j, t_n) &= u_j^n, \\
S_{j+1,n}(x_j, t_n) &= u_{j+1}^n, \\
S_{j,n}^{\left(\frac{1}{2}\right)}(x_j, t_n) &= u_j^{n\left(\frac{1}{2}\right)} = P_j^n, \\
S_{j,n}(x_j, t_n)^{(3)} &= u_j^{n(3)} = M_j^n, \\
S_{j+1,n}(x_j, t_n)^{(3)} &= u_{j+1}^{n\,(3)} = M_{j+1}^n.
\end{aligned}
\tag{6}
$$

According to Caputo fractional derivative for $\left(\frac{1}{2}\right)$ and the conditions (6) substituting into Eq (5), we have:

$$a_j^n = -\frac{\begin{pmatrix} \sqrt{2}e^\theta M_j^n + 2\sqrt{k}\,\cos\,\theta M_j^n - \sqrt{2}M_{1+j}^n \\ -2\sqrt{k}M_{1+j}^n + 2e^\theta k^3 P_j^n - 2k^3\cos\,\theta P_j^n \end{pmatrix}}{\left(\sqrt{2}\cos\,\theta + \sqrt{2}\,\sin\,\theta - \sqrt{2}e^\theta + 2\sqrt{k}\,\sin\,\theta\right)k^3}, \tag{7}$$

$$b_j^n = \frac{\sqrt{2}e^\theta M_j^n - 2\sqrt{k}\,\sin\,\theta M_j^n - \sqrt{2}M_{1+j}^n + 2k^3\,\sin\,\theta P_j^n}{\left(\sqrt{2}\cos\,\theta + \sqrt{2}\,\sin\,\theta - \sqrt{2}e^\theta + 2\sqrt{k}\,\sin\,\theta\right)k^3}, \tag{8}$$

$$c_j^n = \frac{\sqrt{2}\,\sin\,\theta M_j^n + \sqrt{2}\,\cos\,\theta M_j^n - \sqrt{2}M_{1+j}^n + 2k^3\,\sin\,\theta P_j^n}{\left(\sqrt{2}\cos\,\theta + \sqrt{2}\,\sin\,\theta - \sqrt{2}e^\theta + 2\sqrt{k}\,\sin\,\theta\right)k^3}, \tag{9}$$

$$d_j^n = \frac{\begin{pmatrix} \sqrt{2}e^\theta M_j^n - \sqrt{2}\,\cos\,\theta M_j^n + 2\sqrt{k}\,\cos\,\theta M_j^n \\ -2\sqrt{k}\cos\,\theta^2 M_j^n - \sqrt{2}\,\sin\,\theta M_j^n \\ +2\sqrt{2}e^\theta\,\sin\,\theta M_j^n - 2\sqrt{k}\sin\,\theta^2 M_j^n \\ -\sqrt{2}e^\theta M_{j+1}^n - 2\sqrt{k}M_{j+1}^n + \sqrt{2}\,\cos\,\theta M_{j+1}^n \\ +2\sqrt{k}\,\cos\,\theta M_{j+1}^n - \sqrt{2}\,\sin\,\theta M_{j+1}^n \\ +2e^\theta k^3 P_j^n - 2k^3\cos\,\theta P_j^n - 2e^\theta k^3\cos\,\theta P_j^n \\ +2k^3\cos\,\theta^2 P_j^n - 2k^3\,\sin\,\theta P_j^n \\ +2e^\theta k^3\,\sin\,\theta P_j^n + 2k^3\,\sin\,\theta^2 P_j^n - \sqrt{2}e^\theta k^3 u_j^n \\ +\sqrt{2}k^3\cos\,\theta u_j^n + \sqrt{2}k^3\,\sin\,\theta u_j^n + 2k^{\frac{7}{2}}\,\sin\,\theta u_j^n + \sqrt{2}e^\theta k^3 u_{1+j}^n \\ -\sqrt{2}k^3\cos\,\theta u_{1+j}^n - \sqrt{2}k^3\,\sin\,\theta u_{1+j}^n - 2k^{\frac{7}{2}}\,\sin\,\theta u_{1+j}^n \end{pmatrix}}{\left(\sqrt{2}\,\cos\,\theta + \sqrt{2}\,\sin\,\theta - \sqrt{2}e^\theta + 2\sqrt{k}\,\sin\,\theta\right)\theta k^2}, \tag{10}$$

and

$$e_j^n = \frac{\begin{pmatrix} \sqrt{2}e^\theta M_j^n - \sqrt{2}\,\cos\,\theta M_j^n + 2\sqrt{k}\,\cos\,\theta M_j^n \\ -\sqrt{2}\,\sin\,\theta M_j^n - 2\sqrt{k}M_{1+j}^n + 2e^\theta k^3 P_j^n \\ -2k^3\cos\,\theta P_j^n - 2k^3\,\sin\,\theta P_j^n - \sqrt{2}e^\theta k^3 u_j^n \\ +\sqrt{2}k^3\cos\,\theta u_j^n + \sqrt{2}k^3\,\sin\,\theta u_j^n + 2k^{\frac{7}{2}}\,\sin\,\theta u_j^n \end{pmatrix}}{\left(\sqrt{2}\cos\,\theta + \sqrt{2}\,\sin\,\theta - \sqrt{2}e^\theta + 2\sqrt{k}\,\sin\,\theta\right)k^3}. \tag{11}$$

In this context, $\theta = hk$, and the application of the continuity equation $S_{j,n}^{(\tau)}(x_j,\ t_n) = S_{j-1,n}^{(\tau)}(x_j,\ t_n),\ \ \tau = 1, 2$, results in the following expressions:

$$kb_j^n + kc_j^n + d_j^n = -k\ \sin\theta a_{j-1}^n + k\ \cos\ \theta\ b_{j-1}^n + e^\theta k c_{j-1}^n + d_{j-1}^n, \tag{12}$$

$$-k^2 a_j^n + k^2 c_j^n = -k^2\cos\ \theta a_{j-1}^n - k^2\ \sin\theta b_{j-1}^n + e^\theta k^2 c_{j-1}^n. \tag{13}$$

After some simplification and collection, we get:

$$\psi_1 M_{j-1}^n + 2\psi_2 M_j^n + \psi_3 M_{j+1}^n - \Psi(u_{j-1}^n - 2u_j^n + u_{j+1}^n) = \Theta_1 P_j^n - \Theta_2 P_{j-1}^n, \tag{14}$$

$$\Omega_1 M_{j-1}^n - \Omega_2 M_j^n + \Omega_3 M_{j+1}^n = \varrho_1 P_j^n - \varrho_2 P_{j-1}^n, \tag{15}$$

where,

$$\Psi = k^3\left(-\sqrt{2}e^\theta + \sqrt{2}\ \sin\theta + \left(\sqrt{2} + 2\sqrt{k}\right)\ \sin\theta\right),$$

$$\psi_1 = -\left(-2\sqrt{k} + \sqrt{2}(1 + 2e^\theta\theta)\right)\cos\ \theta - \sqrt{2}(1 + 2e^\theta(-1 + \theta))\ \sin\theta$$
$$+ \sqrt{2}e^\theta - 2\sqrt{k},$$

$$\psi_2 = +\left(-\sqrt{2} + \sqrt{2}e^\theta + \left(\sqrt{2} + \sqrt{k}\right)\theta\right)\ \sin\theta$$
$$- 2\sqrt{k} + \left(-\sqrt{k}(-2 + \theta) + \sqrt{2}\theta\right)\cos\ \theta,$$

$$\psi_3 = -\sqrt{2}\ \sin\theta - \sqrt{2}e^\theta - 2\sqrt{k} + 2\sqrt{k}\theta + \left(\sqrt{2} + 2\sqrt{k}\right)\cos\ \theta,$$

$$\Theta_1 = 2(-1 - e^\theta + e^\theta\theta + (1 + e^\theta - \theta)\cos\ \theta - (-1 + e^\theta + \theta)\ \sin\theta)k^3,$$

$$\Theta_2 = 2k^3(-1 - e^\theta + (1 + e^\theta)\cos\ \theta + (1 + e^\theta(-1 + 2\theta))\ \sin\theta),$$

$$\Omega_1 = \sqrt{k} + \sqrt{2}e^\theta\cos\ \theta, \quad \Omega_2 = \sqrt{2}e^\theta + \left(\sqrt{2} + 2\sqrt{k}\right)\cos\ \theta, \quad \Omega_3 = \sqrt{2} + \sqrt{k},$$

$$\varrho_1 = k^3(e^\theta - \cos\ \theta + \ \sin\theta), \ \text{ and }\ \varrho_2 = \ k^3(-1 + e^\theta(\cos\ \theta + \ \sin\theta)).$$

Subtracting Eqs (14) and (15), we have:

$$P_j^n = \frac{\psi_1\varrho_2 - \Theta_2\Omega_1}{\varrho_2\Theta_1 - \Theta_2\varrho_1}M_{j-1}^n + \frac{2\psi_2\varrho_2 + \Theta_2\Omega_2}{\varrho_2\Theta_1 - \Theta_2\varrho_1}M_j^n + \frac{\psi_3\varrho_2 - \Theta_2\Omega_3}{\varrho_2\Theta_1 - \Theta_2\varrho_1}M_{j+1}^n$$
$$- \frac{\Psi\varrho_2}{\varrho_2\Theta_1 - \Theta_2\varrho_1}\left(u_{j-1}^n - 2u_j^n + u_{j+1}^n\right). \tag{16}$$

Upon replacing $j$ in Eq (16) with $j - 1$, we obtain:

$$P_{j-1}^n = \frac{\psi_1\varrho_2 - \Theta_2\Omega_1}{\varrho_2\Theta_1 - \Theta_2\varrho_1}M_{j-2}^n + \frac{2\psi_2\varrho_2 + \Theta_2\Omega_2}{\varrho_2\Theta_1 - \Theta_2\varrho_1}M_{j-1}^n + \frac{\psi_3\varrho_2 - \Theta_2\Omega_3}{\varrho_2\Theta_1 - \Theta_2\varrho_1}M_j^n$$
$$- \frac{\Psi\varrho_2}{\varrho_2\Theta_1 - \Theta_2\varrho_1}\left(u_{j-2}^n - 2u_{j-1}^n + u_j^n\right). \tag{17}$$

Substituting Eqs (16) and (15), into Eq (14), we get:

$$\Xi_1 M_{j-2}^n + \Xi_2 M_{j-1}^n + \Xi_3 M_j^n + \Xi_4 M_{j+1}^n$$
$$= u_{j-2}^n - 2\Upsilon_1 u_{j-1}^n + 2\Upsilon_2 u_j^n - \Upsilon_3 u_{j+1}^n, \tag{18}$$

where,

$$\Xi_1 = \varrho_2 \frac{(\varrho_2\Theta_1 - \Theta_2\varrho_1)(\psi_1\varrho_2 - \Theta_2\Omega_1)}{(\delta_2\Theta_1 - \Theta_2\Theta_1)(\Psi\varrho_2{}^2)},$$

$$\Xi_2 = \eta_1 \frac{\varrho_2\Theta_1 - \Theta_2\varrho_1}{\Psi\varrho_2{}^2} - \varrho_1\frac{\psi_1\varrho_2 - \Theta_2\Omega_1}{\Psi\varrho_2{}^2} + \varrho_2\frac{2\psi_2\varrho_2 + \Theta_2\Omega_2}{\Psi\varrho_2{}^2},$$

$$\Xi_3 = -\Omega_2\frac{\varrho_2\Theta_1 - \Theta_2\varrho_1}{\Psi\varrho_2{}^2} - \varrho_1\frac{2\psi_2\varrho_2 + \Theta_2\Omega_2}{\Psi\varrho_2{}^2} + \varrho_2\frac{\psi_3\varrho_2 - \Theta_2\Omega_3}{\Psi\varrho_2{}^2},$$

$$\Xi_4 = \eta_3\frac{\varrho_2\Theta_1 - \Theta_2\varrho_1}{\Psi\varrho_2{}^2} - \varrho_1\frac{\psi_3\varrho_2 - \Theta_2\Omega_3}{\Psi\varrho_2{}^2},$$

$$\Upsilon_1 = 1 + \frac{\delta_1}{2\varrho_2}, \quad \Upsilon_2 = \frac{1}{2} + \frac{\varrho_1}{\varrho_2}, \quad \text{and} \quad \Upsilon_3 = \frac{\varrho_1}{\varrho_2}.$$

## 3 Truncation error for FNPSM

Consider $T_j$, which characterizes the local truncation error associated with the $j^{th}$ step in the numerical scheme (18). This error gauges the difference between the exact solution of the differential equation at that specific stage and the approximation generated by the method. The analysis of $T_j$ facilitates necessary adjustments to enhance precision. Our principal objective revolves around the minimization of $T_j$, especially as the step size is reduced, ensuring the production of dependable and accurate numerical solutions.

$$
\begin{aligned}
T_j &= u_{j-2}^n - 2\Upsilon_1 u_{j-1}^n + 2\Upsilon_2 u_j^n - \Upsilon_3 u_{j+1}^n \\
&\quad - (\Xi_1 M_{j-2}^n + \Xi_2 M_{j-1}^n + \Xi_3 M_j^n + \Xi_4 M_{j+1}^n) \\
&= u_{j-2}^n - 2\Upsilon_1 u_{j-1}^n + 2\Upsilon_2 u_j^n - \Upsilon_3 u_{j+1}^n \\
&\quad - \Xi_1 M_{j-2}^n - \Xi_2 M_{j-1}^n - \Xi_3 M_j^n - \Xi_4 M_{j+1}^n.
\end{aligned}
\tag{19}
$$

Upon collecting the derivative coefficients following the utilization of Taylor expansion, the ensuing relationship is upheld:

$$
\begin{aligned}
T_j &= (1 - 2\Upsilon_1 + 2\gamma_2 - \Upsilon_3)u_j^n + (-2 + 2\Upsilon_1 - \Upsilon_3)h u_j^{n\prime} \\
&\quad + \left(\frac{4 - 2\Upsilon_1 - \Upsilon_3}{2!}\right)h^2 u_j^n(2) \\
&\quad + \left(\frac{-8 + 2\Upsilon_1 - \Upsilon_3}{3!} + \frac{-\Xi_1 - \Xi_2 - \Xi_3 - \Xi_4}{h^3}\right)h^3 u_j^n(3) \\
&\quad + \left(\frac{16 - 2\Upsilon_1 - \Upsilon_3}{4!} + \frac{2\Xi_1 + \Xi_2 - \Xi_4}{h^3}\right)h^4 u_j^n(4) \\
&\quad + \left(\frac{-32 + 2\Upsilon_1 - \Upsilon_3}{5!} + \frac{-4\Xi_1 - \Xi_2 - \Xi_4}{2!h^3}\right)h^5 u_j^n(5) \\
&\quad + \left(\frac{64 - 2\Upsilon_1 - \Upsilon_3}{6!} + \frac{8\Xi_1 + \Xi_2 - \Xi_4}{3!h^3}\right) + h^6 u_j^n(6) \\
&\quad + \left(\frac{-128 + 2\Upsilon_1 - \Upsilon_3}{7!} + \frac{-16\Xi_1 - \Xi_2 - \Xi_4}{4!\ h^3}\right)h^7 u_j^n(7) \\
&\quad + \left(\frac{256 - 2\Upsilon_1 - \Upsilon_3}{8!} + \frac{32\Xi_1 + \Xi_2 - \Xi_4}{5!h^3}\right) + h^8 u_j^n(8) \\
&\quad + O(k^{2-\alpha} + h^8).
\end{aligned}
\tag{20}
$$

Referring to Eq (19), and setting the coefficient of $u_j^{n(\rho)}$ equal to zero for $\rho = 0, 1, \ldots$, we obtain:

$$\Upsilon_1 = \Upsilon_2 = \frac{3}{2}, \quad \Upsilon_3 = 1, \qquad \Xi_1 = -\frac{h^3}{120}, \Xi_2 = -\frac{19h^3}{40}, \Xi_3 = -\frac{21h^3}{40} \text{ and }, \Xi_4 = \frac{h^3}{120}.$$

Subsequently, the local truncation error, following the substitution of coefficients, is expressed as:

$$T_j = -\frac{119}{160} \ h^8 u_j^n(8) + (2\beta_1 + \beta_2) + O(k^{2-\alpha} + h^8), \tag{21}$$

and the Eq (18) can be written as:

$$u_{j-2}^n - 3u_{j-1}^n + 3u_j^n - u_{j+1}^n = h^3 \left( -\frac{1}{120} M_{j-2}^n - \frac{19}{40} M_{j-1}^n - \frac{21}{40} M_j^n + \frac{1}{120} M_{j+1}^n \right). \tag{22}$$

## 4 Conformable-Caputo fractional non-polynomial spline method

In the forthcoming section, we harness the power of Conformable derivatives along with their inherent properties, seamlessly amalgamating them with the Taylor finite difference scheme. Our objective is to provide a novel solution framework for tackling the time-fractional KdV Eq (1), accompanied by the stipulated initial and boundary conditions (3) and (2). This amalgamation of methodologies holds the promise of unveiling fresh insights into the dynamics of the problem at hand.

The conformable fractional derivative of the function $u(x, t)$ is defined as: [37]

$$\frac{\partial^\alpha u}{\partial t^\alpha} = T_\alpha(u(t)) = \lim_{\omega \to \infty} \frac{u(t + \omega t^{1-\alpha}) - u(t)}{\omega}, \quad 0 < \alpha \le 1. \tag{23}$$

### 4.1 Some properties of conformable fractional derivative

Consider $\alpha \in (0, 1]$, $a \in \mathbb{R}$, and assume $F$ and $G$ are $\alpha$-differentiable at the point $t > 0$, as outlined in [37],

1. $T_\alpha(a_1 F + a_2 G) = a_1 T_\alpha(F) + a_2 T_\alpha(G)$, for all $a_1, a_2 \in \mathbb{R}$.

2. $T_\alpha(t^p) = pt^{p-\alpha}$, for all $p \in \mathbb{R}$.

3. $T_\alpha(F(t)) = 0$, if $F(t)$ is constant function.

4. $T_\alpha(F)(t) = t^{1-\alpha} \frac{df}{dt}(t)$.

5. $T_\alpha(e^{at}) = at^{1-\alpha} e^{at}$.

6. $T_\alpha(\cos at) = -at^{1-\alpha} \sin at$.

7. $T_\alpha(\sin at) = at^{1-\alpha} \cos at$.

Then by finite difference method, we have:

$$\frac{du}{dt} = \frac{u_j^n - u_j^{n-1}}{k}, \quad where \quad u\left(x_j, t_n\right) = u_j^n. \tag{24}$$

According to property (4) in section (4), we have: $T_\alpha(F)(t) = t^{1-\alpha} \frac{df}{dt}(t)$, implies that:

$$\frac{\partial^\alpha u}{\partial t^\alpha} = T_\alpha(u(t)) = t^{1-\alpha} \frac{u_j^n - u_j^{n-1}}{k} = \frac{\mu}{k}\left(u_j^n - u_j^{n-1}\right), \tag{25}$$

where, $\mu = t^{1-\alpha}$, using Eq (1), we have:

$$M_j^n = -\frac{\mu}{k}\left(u_j^n - u_j^{n-1}\right) - \sigma_1 u_j^n \left(\frac{u_j^n - u_{j-1}^n}{h}\right) + f_j^n. \tag{26}$$

Substituting $j$ in Eq (26) with $j-1$, $j-2$, and $j+1$ results in:

$$M_{j-1}^n = -\frac{\mu}{k}\left(u_{j-1}^n - u_{j-1}^{n-1}\right) - \sigma_1 u_{j-1}^n \left(\frac{u_{j-1}^n - u_{j-2}^n}{h}\right) + f_{j-1}^n, \tag{27}$$

$$M_{j-2}^n = -\frac{\mu}{k}\left(u_{j-2}^n - u_{j-2}^{n-1}\right) - \sigma_1 u_{j-2}^n \left(\frac{u_{j-2}^n - u_{j-3}^n}{h}\right) + f_{j-2}^n, \tag{28}$$

and,

$$M_{j+1}^n = -\frac{\mu}{k}\left(u_{j+1}^n - u_{j+1}^{n-1}\right) - \sigma_1 u_{j+1}^n \left(\frac{u_{j+1}^n - u_j^n}{h}\right) + f_{j+1}^n. \tag{29}$$

Substituting Eqs (26)–(29) into Eq (18), we have:

$$
\begin{aligned}
u_{j-2}^n &- 2\Upsilon_1 u_{j-1}^n + 2\Upsilon_2 u_j^n - \Upsilon_3 u_{j+1}^n \\
&= \Xi_1 \left(-\frac{\mu}{k}\left(u_{j-2}^n - u_{j-2}^{n-1}\right) - \sigma_1 u_{j-2}^n \left(\frac{u_{j-2}^n - u_{j-3}^n}{h}\right)\right) \\
&+ \Xi_2 \left(-\frac{\mu}{k}\left(u_{j-1}^n - u_{j-1}^{n-1}\right) - \sigma_1 u_{j-1}^n \left(\frac{u_{j-1}^n - u_{j-2}^n}{h}\right)\right) \\
&+ \Xi_3 \left(-\frac{\mu}{k}\left(u_j^n - u_j^{n-1}\right) - \sigma_1 u_j^n \left(\frac{u_j^n - u_{j-1}^n}{h}\right)\right) \\
&+ \Xi_4 \left(-\frac{\mu}{k}\left(u_{j+1}^n - u_{j+1}^{n-1}\right) - \sigma_1 u_{j+1}^n \left(\frac{u_{j+1}^n - u_j^n}{h}\right)\right) \\
&+ \Xi_1 f_{j-2}^n + \Xi_2 f_{j-1}^n + \Xi_3 f_j^n + \Xi_4 f_{j+1}^n.
\end{aligned}
\tag{30}
$$

Simplifying the implementation, scheme (30) undergoes a transformation into a more user-friendly form, manifesting as the following equation:

$$
\begin{aligned}
A_j \, u_{j-2}^n + B_j \, u_{j-1}^n + C_j \, u_j^n + D_j \, u_{j+1}^n &= A_j^* \, u_{j-2}^{n-1} + B_j^* u_{j-1}^{n-1} \\
&+ C_j^* u_j^{n-1} + D_j^* u_{j+1}^{n-1} + \Xi_1 f_{j-2}^n + \Xi_2 f_{j-1}^n + \Xi_3 f_j^n + \Xi_4 f_{j+1}^n,
\end{aligned}
\tag{31}
$$

where,

$$A_j = 1 + \Xi_1 \frac{\mu}{k} + \Xi_1 \sigma_1 \left( \frac{u_{j-2}^n - u_{j-3}^n}{h} \right),$$

$$B_j = -2\Upsilon_1 + \Xi_2 \frac{\mu}{k} + \Xi_2 \sigma_1 \left( \frac{u_{j-1}^n - u_{j-2}^n}{h} \right),$$

$$C_j = 2\Upsilon_2 + \Xi_3 \frac{\mu}{k} + \Xi_3 \sigma_1 \left( \frac{u_j^n - u_{j-1}^n}{h} \right),$$

$$D_j = -\Upsilon_3 + \Xi_4 \frac{\mu}{k} + \Xi_4 \sigma_1 \left( \frac{u_{j+1}^n - u_j^n}{h} \right),$$

$$A_j^* = \Xi_1 \frac{\mu}{k}, \qquad B_j^* = \Xi_2 \frac{\mu}{k}, \qquad C_j^* = \Xi_3 \frac{\mu}{k}, \qquad \text{and} \qquad D_j^* = \Xi_4 \frac{\mu}{k}.$$

## 5 Stability analysis for conformable-Caputo time-fractional KdV equation

In the realm of Von Neumann stability analysis, a foundational assumption is posited regarding the solution structure of the Eq (31). This assumption delineates the form of the solution in the following manner:

$$u_j^n = \Psi^n e^{i\,\epsilon\,j\,h}. \tag{32}$$

Here, $\epsilon$ denotes the real spatial wave number, and $i = \sqrt{-1}$. By transforming the nonlinear term into a linear expression and substituting Eq (32) into Eq (30), we can assess the stability of the proposed method for solving Eq (1). This leads to:

$$\Psi^n \left( e^{i\,\epsilon\,h\,j} e^{-2i\,\epsilon\,h} - 3e^{i\,\epsilon\,h\,j} e^{-i\,\epsilon\,h} + 3e^{i\,\epsilon\,j\,h} - e^{i\,\epsilon\,h\,j} e^{i\,\epsilon\,h} \right)$$

$$-\Psi^n \left( \frac{\mu h^3}{120k} + \frac{\sigma_1 d h^3}{120} \right) \left( e^{i\,\epsilon\,h\,j} e^{-2i\,\epsilon\,h} + 57e^{i\,\epsilon\,h\,j} e^{-i\,\epsilon\,h} + 63e^{i\,\epsilon\,j\,h} - e^{i\,\epsilon\,h\,j} e^{i\,\epsilon\,h} \right) \tag{33}$$

$$= -\Psi^{n-1} \frac{\mu h^3}{120k} \left( e^{i\,\epsilon\,h\,j} e^{-2i\,\epsilon\,h} + 57e^{i\,\epsilon\,h\,j} e^{-i\,\epsilon\,h} + 63e^{i\,\epsilon\,j\,h} - e^{i\,\epsilon\,h\,j} e^{i\,\epsilon\,h} \right).$$

Collecting terms after some simplification, we have:

$$\Psi^n \left( \frac{(e^{-2i\,\epsilon\,h} - 3e^{-i\,\epsilon\,h} + 3 - e^{i\,\epsilon\,h})}{(e^{-2i\,\epsilon\,h} + 57e^{-i\,\epsilon\,h} + 63 - e^{i\,\epsilon\,h})} - \left( \frac{\mu h^3}{120k} + \frac{\sigma_1 d h^3}{120} \right) \right)$$

$$= -\Psi^{n-1} \frac{\mu h^3}{120k}. \tag{34}$$

Applying Euler's formula $e^{i\,\varrho} = \cos\varrho + i\,\sin\varrho$, $\varrho = h\epsilon$, Eq (34) transforms into:

$$\Psi^n \frac{120}{h^3} \left( \frac{A^* + iB^*}{A + iB} - \left( \frac{\mu}{k} + \sigma_1 d \right) \right) = -\Psi^{n-1} \frac{\mu}{k}, \tag{35}$$

where,

$$A^* = \cos 2\varrho - 4\cos\varrho + 3, \qquad A = \cos 2\varrho + 56\cos\varrho + 63,$$

$$B^* = -\sin 2\varrho + 2\sin\varrho, \qquad B = -\sin 2\varrho - 58\sin\varrho.$$

Then

$$\Psi^n = -\Psi^{n-1} \frac{\left(\frac{\mu}{k} + \sigma_1 d\right)}{\varpi} , \qquad (36)$$

where $\varpi = \left(\frac{A^* + iB^*}{A + iB} - \left(\frac{\mu}{k} + \sigma_1 d\right)\right)$. Using the norm of both sides, we have:

$$|\Psi^n| = \left|-\Psi^{n-1} \frac{\left(\frac{\mu}{k} + \sigma_1 d\right)}{\varpi}\right| \leq |\Psi^{n-1}|, \qquad n = 1, 2, \dots, \ N-1. \qquad (37)$$

This leads to:

$$|\Psi^n| \leq |\Psi^0|. \qquad (38)$$

Consequently, the developed method demonstrates stability without any imposed conditions, rendering it unconditionally stable.

## 6 Numerical illustration and discussion of conformable-Caputo non-polynomial spline method

In this particular section, we employ the fractional non-polynomial spline methodology for the resolution of the fractional Korteweg-de Vries problem (1), wherein both Caputo and conformable derivatives are considered within the framework of fractional conditions. A comprehensive comparative analysis is conducted between the outcomes derived from our fractional non-polynomial spline method (FNPSM) and the exact solution, aimed at elucidating the accuracy and efficacy inherent in the proposed computational paradigm. The exposition of findings is facilitated through the presentation of illustrative figures and tabulated data, generated using MATLAB software. Additionally, the maximum absolute errors and least square errors are calculated and compared to well-known values:

$$L_\infty = \max_{1 \leq j \leq M} |u_{j_{exact}} - u_{j_{numerical}}|, \qquad (39)$$

and

$$L_2 = \sqrt{h \sum_{i=1}^{M} |u_{j_{exact}} - u_{j_{numerical}}|^2}. \qquad (40)$$

### 6.1 Example 1

Examine the time-fractional Korteweg-de Vries [22], where $\sigma_1 = 6$, in Eq (1) given as:

$$\frac{\partial^\alpha u}{\partial t^\alpha} + \sigma_1 u \frac{\partial u}{\partial x} + \frac{\partial^3 u}{\partial x^3} = F(x, t), \qquad 0 < \alpha \leq 1, \qquad t \geq 0, \qquad 0 \leq x \leq 1, \qquad (41)$$

with the initial condition:

$$u(x, 0) = x(1 - x)^2 \qquad 0 \leq x \leq 1, \qquad (42)$$

and the boundary conditions:

$$u(0, t) = u(1, t) = u_x(1, t) = 0 \qquad 0 < t \leq 1, \qquad (43)$$

and

$$F(x,t) = \frac{-\Gamma(7)}{\Gamma(7-\alpha)} t^{6-\alpha} \; x(1-x)^2 + \sigma_1(1-t^6)^2$$
$$+ (3x^5 - 10x^4 + 12x^3 - 6x^2 + x) + 6\sigma_2(1-t^6). \tag{44}$$

and exact solution analytically is $u(x, t) = (1 - t^6)x(1 - x)^2$.

This study embarked on a comprehensive exploration of numerical approaches, employing the Conformable-Caputo FNPSM method, to address the fractional Korteweg-de Vries Eq (41) under specified conditions (42)–(44). The results of our comparative analysis, depicted in Fig 1, underscore a significant convergence between analytical and numerical solutions. This alignment serves as a robust indicator of the elevated precision inherent in our chosen numerical approach, affirming its methodological soundness and efficacy. Expanding the application of the Conformable-Caputo FNPSM methodology, three-dimensional depictions of the solution $u(x, t)$ for **Example 1** are presented in Fig 2. These depictions vividly portray the evolving characteristics of the solutions within the prescribed parameter domain $0 \leq x$, $t \leq 1$, while maintaining a fixed fractional order at $\alpha = 0.75$. Exploring the influence of the fractional order $\alpha$ on the solution $u(x, t)$ for **Example 1**, as depicted in Fig 3, our observations indicate a negligible impact of the fractional order on $u(x, t)$ in proximal regions of approximately $x = 0.38$. However, as $\alpha = 1.0$, a discernible augmentation is observed. Conversely, beyond $x = 0.38$, an inverse relationship is noted, where an increase in $\alpha$ correlates with a decrease in the value of $u(x, t)$. To elucidate the temporal influence on the solution $u(x, t)$ for **Example 1**, Conformable-Caputo FNPSM was employed, as illustrated in Fig 4. Maintaining a constant $\alpha = 0.25$ within the domain $0 \leq x \leq 1$, our observations discern temporal alterations in the

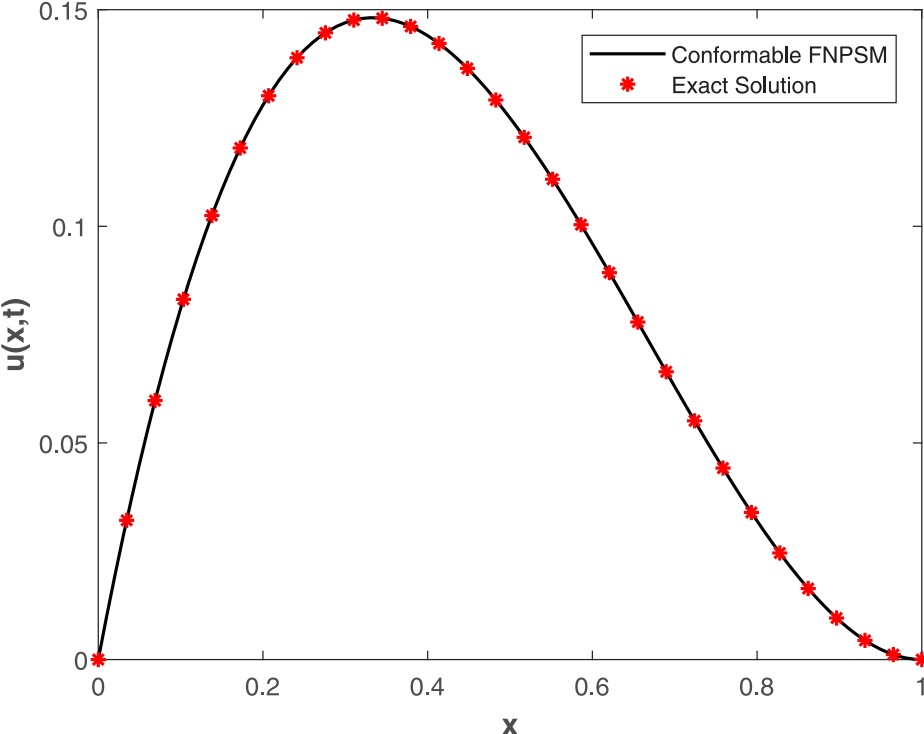

**Fig 1. Curve of exact and conformable-Caputo FNPS solutions for Example 1, where $x \in [0, 1]$, $t = 0.2$ and $\alpha = 0.5$.**

**Conformable Fractional Non-polynomial Spline Method**

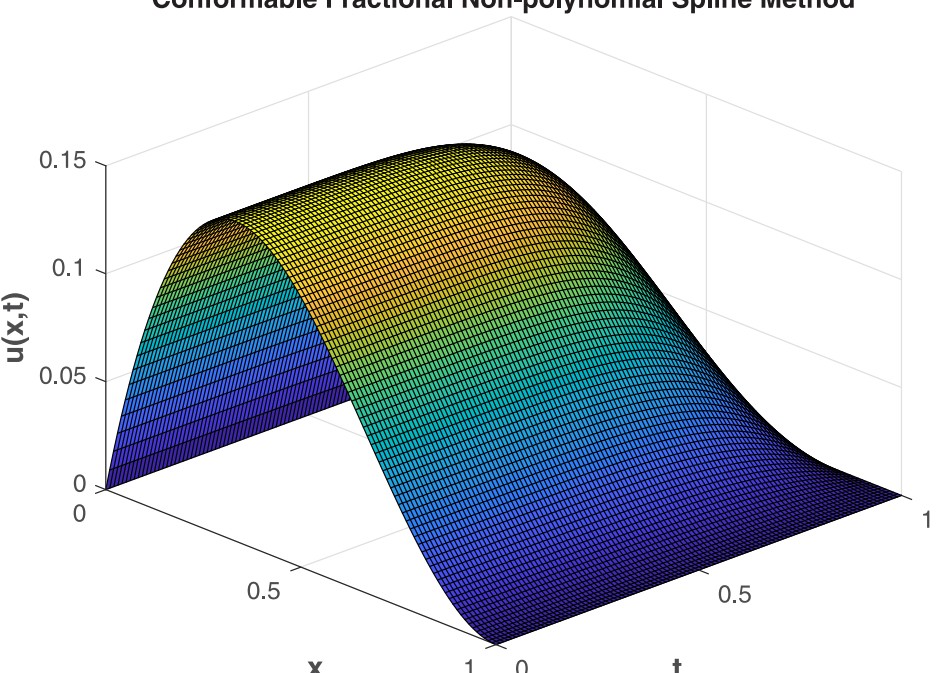

**Fig 2. 3D mesh plot of numerical solution for Example 1, where** $x \in [0, 1]$ $t \in [0, 1]$ **and** $\alpha = 0.75$.

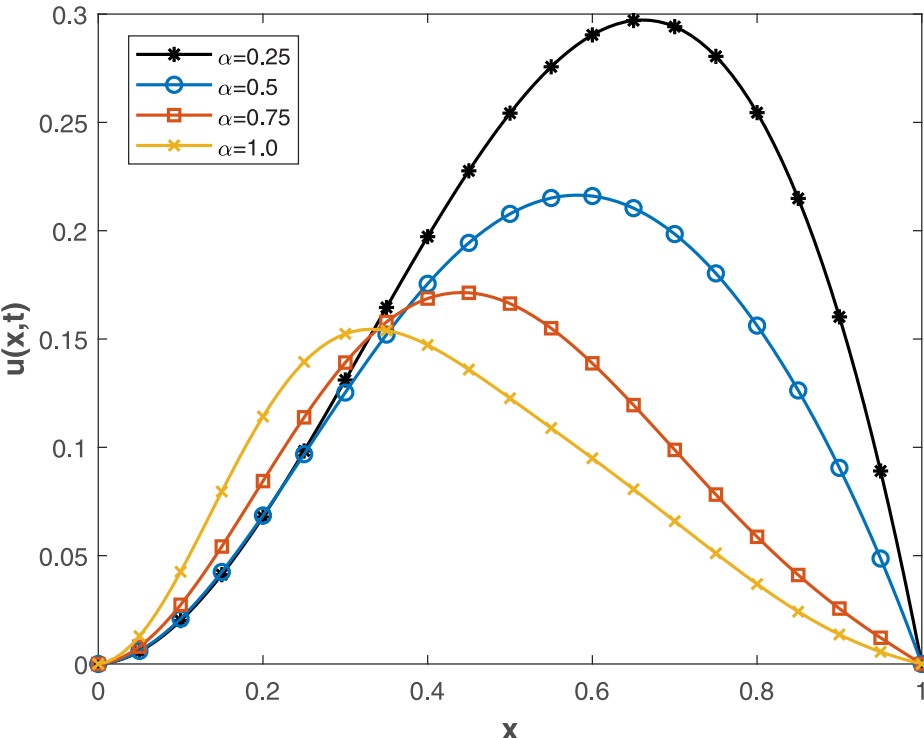

**Fig 3. Fractional order** $\alpha$ **effect on** $u(x, t)$ **for Example 1, where** $0 \leq x \leq 1$ **and** $t = 0.2$.

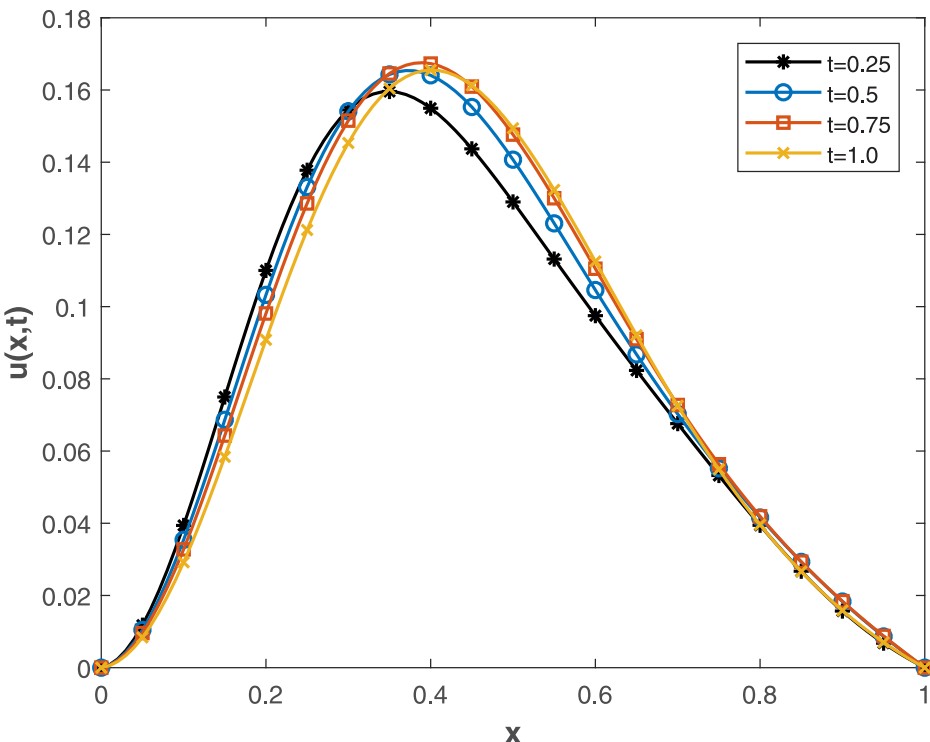

**Fig 4. Time different value effect on $u(x, t)$ for Example 1, where $\alpha = 0.25$ and $0 \leq x \leq 1$.**

value of $u(x, t)$ solely within the region $0.2 < x < 0.75$. Notably, an increase in time corresponds to an augmentation in the value of $u(x, t)$. In addition to these visual representations, a meticulous assessment of performance norm errors between analytical and numerical solutions was conducted using Conformable-Caputo FNPSM for **Example 1**. The findings outlined in Table 1 demonstrate the maximum norm error comparison between our method and the Finite difference/collocation method (FDCM) [38], clearly indicating the superiority of our presented method. Furthermore, norm error comparison in Table 2 reinforces the suitability of our method, affirming its effectiveness and superior results.

### 6.2 Example 2

Examine the time fractional Korteweg-de Vries [39], where $\sigma_1 = 6$ and $F(x, t) = 0$, in Eq (1) given as:

$$\frac{\partial^\alpha u}{\partial t^\alpha} + \sigma_1 u \frac{\partial u}{\partial x} + \frac{\partial^3 u}{\partial x^3} = F(x, t), \qquad 0 < \alpha \leq 1, \qquad t \geq 0, \qquad -10 \leq x \leq 10, \qquad (45)$$

**Table 1. Maximum absolute error comparison between conformable-Caputo FNPSM and finite difference/collocation method (FDCM) [38] for Example 1, where $x \in [0, 1]$.**

| | $\alpha = 0.2$ | | $\alpha = 0.9$ | |
|---|---|---|---|---|
| $N$ | CCFNPSM | FDCM | CCFNPSM | FDCM |
| 100 | $1.0446 \times 10^{-07}$ | $1.39 \times 10^{-06}$ | $2.2972 \times 10^{-06}$ | $1.71 \times 10^{-04}$ |
| 200 | $3.2321 \times 10^{-07}$ | $4.28 \times 10^{-07}$ | $3.4765 \times 10^{-05}$ | $8.04 \times 10^{-05}$ |
| 300 | $2.0212 \times 10^{-07}$ | $2.13 \times 10^{-07}$ | $2.2175 \times 10^{-05}$ | $5.16 \times 10^{-05}$ |
| 400 | $3.4356 \times 10^{-08}$ | $1.29 \times 10^{-07}$ | $2.9352 \times 10^{-06}$ | $3.76 \times 10^{-05}$ |

**Table 2. Error norm comparison between conformable-Caputo FNPSM and exact solution for Example 1, where $x \in [0, 1]$ and $\alpha = 0.5$.**

| | Conformable-Caputo FNPSM | |
|---|---|---|
| $t$ | $L_\infty$ | $L_2$ |
| 0.02 | $5.2432 \times 10^{-06}$ | $3.2168 \times 10^{-06}$ |
| 0.04 | $3.2365 \times 10^{-05}$ | $4.5468 \times 10^{-06}$ |
| 0.06 | $2.3632 \times 10^{-05}$ | $2.2197 \times 10^{-06}$ |
| 0.08 | $1.9854 \times 10^{-06}$ | $3.0167 \times 10^{-06}$ |

with the exact solution, where $\alpha = 0.5$ as follows:

$$u(x, t) = \frac{1}{2} \operatorname{sech}^2 \left( \frac{1}{2} (x - t) \right). \tag{46}$$

Our study delves deep into the intricacies of numerical approaches, employing the Conformable-Caputo FNPSM method to address the fractional Korteweg-de Vries equation under specific conditions derived from the exact solution. The visual representations in Fig 5 provide a clear insight into the convergence between analytical and numerical solutions, emphasizing the precision of the chosen numerical approach. Expanding the application of Conformable-Caputo FNPSM to create three-dimensional depictions for **Example 2**, showcased in Fig 6, adds depth to your research by offering a vivid portrayal of the evolving characteristics of the solution. This meticulous exploration within the parameter domain, coupled with a fixed fractional order at $\alpha = 0.5$, enhances the comprehensiveness of your investigation. Moving

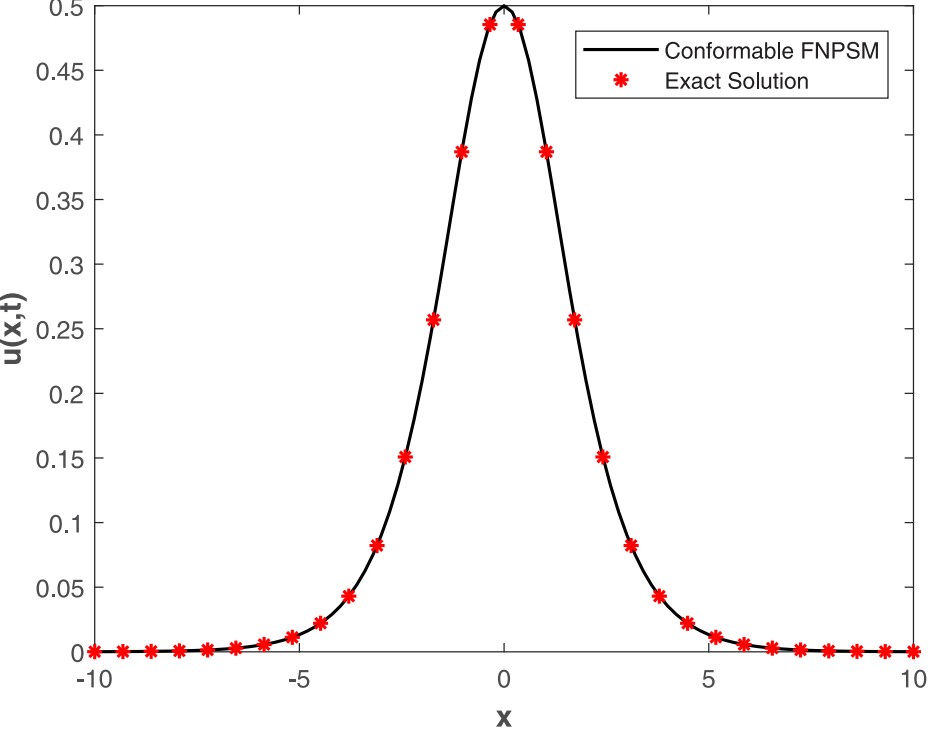

**Fig 5. Curve of exact and conformable-Caputo FNPS solutions for Example 2, where $x \in [0, 1]$, $t = 0.2$ and $\alpha = 0.5$.**

**Conformable Fractional Non-polynomial Spline Method**

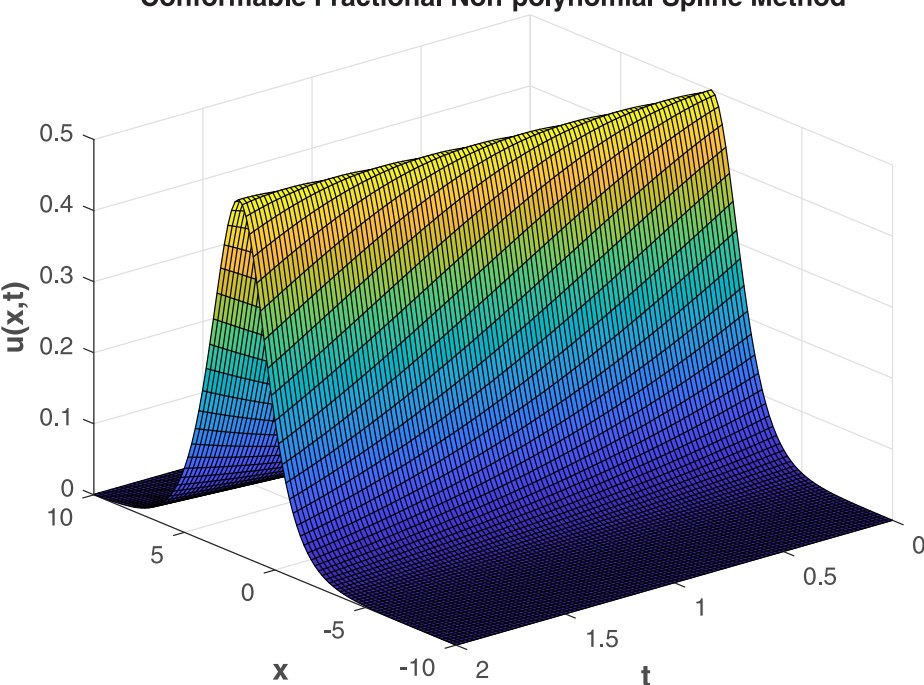

**Fig 6. 3D mesh plot of numerical solution for Example 2, where $x \in [0, 1]$ $t \in [0, 1]$ and $\alpha$ = 0.75.**

forward, your study systematically delves into the temporal dynamics governing the solution $u(x, t)$ for **Example 2**. Employing the rigorous methodology of Conformable-Caputo FNPSM, delineated in Fig 7, the steadfast maintenance of $\alpha$ = 0.5 within the expansive domain $-10 \leq x \leq 10$ establishes a robust foundation for your observations. Notably, temporal fluctuations in the value of $u(x, t)$ are discerned exclusively within the localized region of $-6.0 < x < 6.0$, with a conspicuous augmentation corresponding to the progression of time. Supplementing this visual exploration, your methodical evaluation of performance norm errors, meticulously documented in Table 3, provides a comprehensive norm error comparison between your method and the exact solution.

## 7 Conclusion

In conclusion, our investigation into the time-fractional Korteweg-de Vries equation has introduced a significant advancement in numerical methodologies with the conformable-Caputo fractional non-polynomial spline method. This innovative approach, rooted in the fusion of conformable and Caputo fractional calculus within a non-polynomial spline framework, overcomes the challenges associated with the complex behavior of the KdV equation. Our method offers precision and modeling capabilities, validated through graphical representations and comparative analysis. The stability analysis establishes the robustness of the proposed approach within a specific parameter range. Despite acknowledging limitations, our research contributes to the field of numerical analysis and algorithm development. The conformable-Caputo fractional non-polynomial spline method emerges as a versatile tool for modeling, offering potential applications beyond the time-fractional KdV equation. In essence, this study provides a robust solution to the challenges posed by the time-fractional KdV

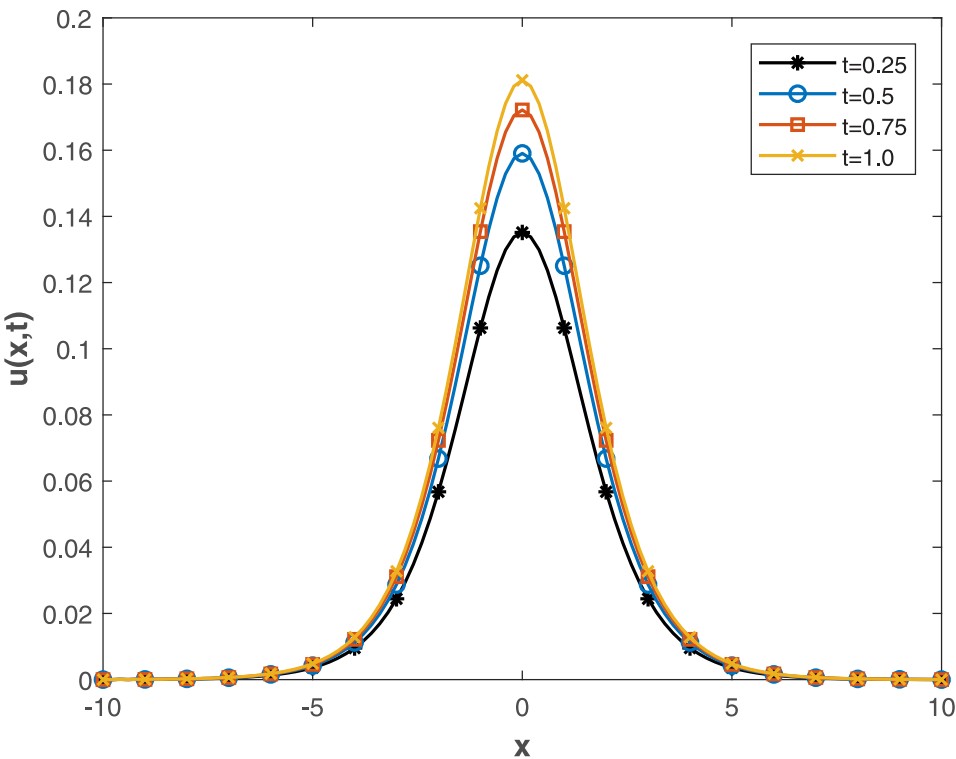

**Fig 7. Time different value effect on $u(x, t)$ for Example 2, where $\alpha$ = 0.25 and $0 \leq x \leq 1$.**

**Table 3. Error norm comparison between Caputo FNPSM and conformable FNPSM for Example 2, where $x \in [0, 1]$ and $\alpha$ = 0.5.**

| Conformable-Caputo FNPSM | | |
|---|---|---|
| $t$ | $L_\infty$ | $L_2$ |
| 0.02 | $6.4376 \times 10^{-07}$ | $6.4285 \times 10^{-06}$ |
| 0.04 | $4.2374 \times 10^{-06}$ | $5.3174 \times 10^{-05}$ |
| 0.06 | $4.3264 \times 10^{-06}$ | $3.4279 \times 10^{-05}$ |
| 0.08 | $5.4322 \times 10^{-06}$ | $4.6015 \times 10^{-06}$ |

equation and illuminates new possibilities for accurate simulations in scientific disciplines such as fluid mechanics, plasma physics, and mathematical physics.

## Author Contributions

**Conceptualization:** Majeed A. Yousif, Mohamed Abdelwahed.

**Data curation:** Majeed A. Yousif.

**Formal analysis:** Ahmad Zeeshan, Mohamed Abdelwahed.

**Funding acquisition:** Mohamed Abdelwahed.

**Investigation:** Majeed A. Yousif, Mohamed Abdelwahed.

**Methodology:** Majeed A. Yousif.

**Project administration:** Majeed A. Yousif, Faraidun K. Hamasalh, Mohamed Abdelwahed.

**Resources:** Majeed A. Yousif, Faraidun K. Hamasalh, Ahmad Zeeshan.

**Software:** Majeed A. Yousif, Ahmad Zeeshan.

**Supervision:** Faraidun K. Hamasalh, Mohamed Abdelwahed.

**Validation:** Majeed A. Yousif, Ahmad Zeeshan.

**Visualization:** Ahmad Zeeshan.

**Writing – original draft:** Majeed A. Yousif.

**Writing – review & editing:** Faraidun K. Hamasalh, Ahmad Zeeshan, Mohamed Abdelwahed.

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
