## [Decision Letter · Decision Letter 0]

15 Apr 2024

PONE-D-24-12081Efficient Simulation of Time-Fractional KdV Equation via Conformable-Caputo Non-Polynomial Spline MethodPLOS ONE

Dear Dr. Yousif,

Thank you for submitting your manuscript to PLOS ONE. After careful consideration, we feel that it has merit but does not fully meet PLOS ONE’s publication criteria as it currently stands. Therefore, we invite you to submit a revised version of the manuscript that addresses the points raised during the review process.

**Dear Authors,**

**Revised the paper carefully and keep the following things in your mind:**

**1- Minimize the use of irrelevant citations. Authors whose submitted manuscripts are found to include citations whose primary purpose is to increase the number of citations to a given author’s work, or to articles published in a particular journal, may incur sanctions.**

**2- Manuscripts found to contain citations primarily included to boost the citation count of a specific author's work or articles from a particular journal may face penalties. If you are citing papers recommended by the reviewers, it is important to justify why these papers have been included and how they are relevant to your study. It is advisable to limit the inclusion of reviewer-suggested papers to a maximum of two.**

We look forward to receiving your revised manuscript.

Kind regards,

Muhammad Mubashir Bhatti

Academic Editor

PLOS ONE

Additional Editor Comments:

**Dear Authors,**

**Revised the paper carefully and keep the following things in your mind:**

**1- Minimize the use of irrelevant citations. Authors whose submitted manuscripts are found to include citations whose primary purpose is to increase the number of citations to a given author’s work, or to articles published in a particular journal, may incur sanctions.**

**2- Manuscripts found to contain citations primarily included to boost the citation count of a specific author's work or articles from a particular journal may face penalties. If you are citing papers recommended by the reviewers, it is important to justify why these papers have been included and how they are relevant to your study. It is advisable to limit the inclusion of reviewer-suggested papers to a maximum of two.**

Reviewers' comments:

Reviewer's Responses to Questions

**Comments to the Author**

1. Is the manuscript technically sound, and do the data support the conclusions?

Reviewer #1: Partly

Reviewer #2: Yes

2. Has the statistical analysis been performed appropriately and rigorously? 

Reviewer #1: Yes

Reviewer #2: N/A

3. Have the authors made all data underlying the findings in their manuscript fully available?

Reviewer #1: Yes

Reviewer #2: Yes

4. Is the manuscript presented in an intelligible fashion and written in standard English?

Reviewer #1: Yes

Reviewer #2: Yes

5. Review Comments to the Author

Reviewer #1: This paper is concerned with the conformable-Caputo fractional non-polynomial spline method for solving the time-fractional Korteweg-de Vries equation. Von Neumann approach demonstrates unconditional stability of the scheme, and its validity is shown. Moreover, quantitative assessment using L2 and L∞ error norms confirms its superiority, with worked examples.

The presented results are in current area of research and their appearance in this platform would serve good to the scientific community. In conclusion, this paper is acceptable to appear in Plos One, provided that the subsequent modifications are accounted for in a possible revision:

A. The language is good overall, requiring some fix in typos and grammars.

B. Introduction could be more informative in terms of novelty and findings.

C. Differentiate the current work from the recent published (spline methods).

D. The model in (1-3) should be referenced properly from the literature.

E. The Caputo Derivative definition should be given. Why do you call the method Conformable-Caputo?

Caputo defition is given in terms of integrals, but Eq.(22) is not!

F. The literature seems weak and not complete missing the latest publications, such as https://doi.org/10.1016/j.jocs.2022.101841, https://doi.org/10.1108/HFF-04-2022-0262, https://doi.org/10.1016/j.chaos.2022.112980, and DOI: 10.32604/cmes.2022.020781.

G. Why is the developed method rendering unconditionally stable?

H. Examples chosen are simple and having exact solutions, without dependance on alpha derivative order. Test your method on a more complex KdV model without analytical solution.

K. The Conclusion should be trimmed with focus on the novelty and innovativeness.

L. Comparisons with other methods of the conformable-Caputo fractional non-polynomial spline method is not fulfilled.

Reviewer #2: Before the Editor makes a decision, I suggest that the authors must take into account the following corrections:

1. I don't think the abbreviations used in the title are inspired.

2. The "Introduction" section should be more concise.

3. It is not usual for mathematical relations to appear in the Introduction section.

4. Origin of equation (1) is not specified.

5. What is the motivation of the relations (4)?

6. The expressions for d_j and e_j in (9) and (10) are “bushy”. It is difficult to check their correctness.

7. Details on obtaining relation (19) are required.

8. It is not clear how were obtained the data in Tables.

9. From where were taken the data used in the graphic representations?

10. Some editing "glitches" need to be corrected.

11. References are not uniformly written. In some references the name of the journal is written in full and in other it is abbreviated.

12. Also, I think, the authors must strengthen the References section with some articles that use some similar techniques, to make the techniques used more plausible, for instance: On mixed problem in thermos-elasticity of type III for Cosserat media, Journal of Taibah University for Science, 2022; 16(1): 1264–1274; The Effects of Fractional Time Derivatives in Porothermoelastic Materials Using Finite Element Method, Mathematics. 2021; 9(14): Art. No. 1606; An evolutionary equation in thermoelasticity of dipolar bodies, Journal of Mathematical Physics, 1999; 40(3): 1391-1399.

If the authors take into account all these corrections, then this manuscript deserves to be published.

6. PLOS authors have the option to publish the peer review history of their article (what does this mean?). If published, this will include your full peer review and any attached files.

Reviewer #1: No

Reviewer #2: No

---

## [Author Response · Author response to Decision Letter 0]

26 Apr 2024

Dear Editor, 

About citations, we only used relevant citations.

About reviewer comments, we uploaded a file "Response to Referees" and answered all comments requested point by point.

Sincerely...

---

## [Decision Letter · Decision Letter 1]

1 May 2024

Efficient Simulation of Time-Fractional Korteweg-deVries Equation via Conformable-Caputo Non-Polynomial Spline Method

PONE-D-24-12081R1

Dear Dr. Yousif,

We’re pleased to inform you that your manuscript has been judged scientifically suitable for publication and will be formally accepted for publication once it meets all outstanding technical requirements.

Kind regards,

Muhammad Mubashir Bhatti

Academic Editor

PLOS ONE

Additional Editor Comments (optional):

Reviewers' comments:

Reviewer's Responses to Questions

**Comments to the Author**

1. If the authors have adequately addressed your comments raised in a previous round of review and you feel that this manuscript is now acceptable for publication, you may indicate that here to bypass the “Comments to the Author” section, enter your conflict of interest statement in the “Confidential to Editor” section, and submit your "Accept" recommendation.

Reviewer #1: (No Response)

Reviewer #2: All comments have been addressed

2. Is the manuscript technically sound, and do the data support the conclusions?

Reviewer #1: Yes

Reviewer #2: Yes

3. Has the statistical analysis been performed appropriately and rigorously? 

Reviewer #1: Yes

Reviewer #2: Yes

4. Have the authors made all data underlying the findings in their manuscript fully available?

Reviewer #1: Yes

Reviewer #2: Yes

5. Is the manuscript presented in an intelligible fashion and written in standard English?

Reviewer #1: Yes

Reviewer #2: Yes

6. Review Comments to the Author

Reviewer #1: Proper revision has been fulfilled. Accept now as it is. Proper revision has been fulfilled. Accept now.

Reviewer #2: The authors took into account all the corrections I suggested, which led to an improved form of the manuscript.

7. PLOS authors have the option to publish the peer review history of their article (what does this mean?). If published, this will include your full peer review and any attached files.

Reviewer #1: No

Reviewer #2: No

---

## [Editor Report · Acceptance letter]

14 Jun 2024

PONE-D-24-12081R1 

PLOS ONE

Dear Dr. Yousif, 

I'm pleased to inform you that your manuscript has been deemed suitable for publication in PLOS ONE. Congratulations! Your manuscript is now being handed over to our production team.

Kind regards, 

on behalf of

Dr. Muhammad Mubashir Bhatti 

Academic Editor

PLOS ONE